# Gender Differences in Cyberstalking: The Roles of Risk, Control, and Opportunity Factors in Social Media

**DOI:** 10.3390/bs15050566

**Published:** 2025-04-22

**Authors:** Seong-Sik Lee, Cheong Sun Park

**Affiliations:** 1Department of Information Sociology, Soongsil University, Seoul 06978, Republic of Korea; lss824@ssu.ac.kr; 2Department of Public Administration, Korean National Police University, Asan 31539, Republic of Korea

**Keywords:** cyberstalking, risk factors, control factors, opportunity factors, gender differences

## Abstract

This study empirically tests explanatory factors for cyberstalking on social networking services (SNS), especially focusing on gender differences in the effects of risk, control, and opportunity factors. In this study, we used lack of attachment and denial of victim as risk factors, morality and self-control as control factors, and anonymity as an opportunity factor. We predicted that the main risk effect for cyberstalking and the interaction effect between risk and control factors and between risk and opportunity factors can be differentiated by gender. It is hypothesized that the effects of lack of attachment and denial of victim as risk factors for cyberstalking would differ by gender. Furthermore, in the context of risk factors, we predicted that the moderating effect of the control factor would be greater for women, and the effect of the opportunity factor, such as anonymity, would be greater for men. The results of the analysis of cross-sectional data from 270 SNS using college students in Seoul, South Korea, generally supported the hypotheses. As a risk factor, the influence of lack of attachment was greater for men, while denial of victim was greater for women. The moderating effects of the control factors were greater for women in such a way that the interaction between denial of victim and morality was significant for women; while the moderating effect of the opportunity factor was greater for men in such a way that the interaction between lack of attachment and anonymity was more significant for men. This study finds that the risk factors of cyberstalking and the respective moderating effects of control and opportunity factors can be differentiated according to gender.

## 1. Introduction

Stalking refers to the act or crime of intentionally and repeatedly following or harassing another person in circumstances that would cause a reasonable person to fear injury or death due to the threat expressed or implied. Cyberstalking is the same act or crime performed through information and communication networks, and there is no significant difference between the two definitions other than the location of the stalking ([7]). Cyberstalking began to emerge as one of the main issues studied by criminologists, who duly described the damage of cyberstalking in the early days ([6]; [56]). However, scholars have shifted their attention to the causality of perpetration with the help of traditional sociological theories, such as strain theory, social learning theory, and self-control theory ([15]; [22]; [43]; [67]), or psychological factors, such as hostility, jealousy, and attachment problems ([18]; [64]). Even so, empirical research on cyberstalking is still lacking, and more evidence is needed.

This study explores the explanatory factors for cyberstalking on an SNS (social networking service), especially focusing on gender differences among college students. According to the 2020 Cyber Violence survey conducted by the National Information Society Agency (NIA)[note 1] in South Korea, only 0.8% of youths surveyed had offending experiences of cyberstalking, while this figure was 19.1% for adults. In this regard, this study intends to examine the cyberstalking of adults, especially university students, who have relatively higher offense experiences of cyberstalking. In addition, this study examines whether the explanatory factors for cyberstalking are differentiated by gender.

Previous studies have suggested that men are often the perpetrators of stalking, while women are the victims. Furthermore, there are many cases in which the victim is an ex-girlfriend or lover when the perpetrator is male, whereas for female perpetrators, the victim is a same-gender friend ([49]; [60]). It has also been argued that the motivation for a male stalker is a high level of obsession or retaliation for the opposite gender who informed him of a break-up; while the motivation for female stalkers is an extension of bullying by same-gender peers ([54]). In addition, stalking by men is more violent than stalking by women, and is more likely to include physical violence ([32]; [57]). Based on these characteristics, it can be presumed that the causes of stalking between men and women are somewhat different. This study analyzes whether the motives and causes of cyberstalking differ between men and women. Unlike traditional offline stalking, perpetrators and victims in cyberstalking often do not know each other, and the stereotypical role assignment of male perpetrators and female victims is not necessarily the case (or applicable) anymore in cyberstalking ([9]; [18]). Nevertheless, cyberstalking studies ([19]) have shown that, although the rate of female perpetrators has increased compared to conventional stalking, it is still a norm that males are perpetrators and females are victims, and the target is often former girlfriends or lovers.

In this study, we used various explanatory factors for SNS stalking and attempted to verify it. The existing factors can be divided into risk, control, and opportunity factors, and the interactive mechanism of these three factors is examined. The risk factor refers to the motive and causative factor that induces cyberstalking, the control factor is the factor that prevents cyberstalking, and, conversely, the opportunity factor is the situational opportunity factor for the occurrence of stalking. This study examines the main risk effect on cyberstalking and the interaction effect between the risk and control factors and between the risk and opportunity factors.

The most notable aspect of this study is the application of such interaction effects to men and women separately, and the examination of the gender differences in the results. Previous studies have shown gender differences in the causes of conventional stalking ([54]; [60]): the research question is whether gender differences exist in the main effects and interaction effects of the three factors—risk, control, and opportunity—in cyberstalking. We will examine whether men and women differ in their risk, control, and opportunity factors for cyberstalking and explore the implications and alternatives from the results.

## 2. Literature Review

### 2.1. Risk Factors in Cyberstalking and Gender Differences

Stalking can be classified into three types: simple obsession, affectionate obsession, and delusion ([70]). Simple obsession is derived from a form of attachment or the restoration of a relationship with a former spouse or lover, while affectionate obsession arises from an expression of affection for someone they love. The delusional type of stalking arises from the illusion (false belief) that the victim loves the perpetrator. In the case of cyberstalking, apart from obsession and delusion, retaliation against the victim is also proposed as a major of offense ([50]). [46] ([46]) classified cyberstalking into four types, including obsession, retaliation, harassment, and grouping.

As we see in the cases of simple obsession and affection obsession, the main forms of stalking begin with the so-called obsession. Obsession is a major factor in men’s stalking behavior toward their former lovers, and is known to be caused by a lack of attachment. It has been argued that stalking arises out of confusion in attachment between the actor and his/her parents during childhood. Previous studies on attachment problems have shown that attachment with parents in childhood is very important, and lack of affection or violence from the parents is more likely to manifest in the form of preoccupied, anxious, and insecure attachments rather than stable attachments. These characteristics of abnormal attachment are prevalent among stalkers suffering from relational problems ([16]; [20]; [52]). Individuals with attachment disorders want to control others in the relationship, even in coercive ways, because they want to receive attention and reassurance from them, which leads to an increased outbreak of stalking. Moreover, fear of rejection, suspicion of others, and probable breakup of intimate relationships tend to produce anger and subsequent violence in extreme cases.

In particular, heterosexual men with insecure attachment disorders tend to be obsessive with their partners and try to dominate them, which leads to stalking. Although few stalking studies have applied attachment discussion to men and women separately, dating violence studies show that, unlike women, men often try to control their partner in their relationship with ex-partners, and often use violence to vent their resentment against separation. Therefore, if male violence is somehow related to attachment problems ([23]), and if cyberstalking is an extension of offline stalking, then attachment is a more suitable predictor for male stalking as compared to female stalking.

It has also been argued that cyberstalking is not just a matter of affection and obsession, but is rather motivated by violent relationships, such as bullying, retaliation, and punishment against others ([62]). Cyberstalking is very similar to cyberbullying, as it is a form of bullying (or violence) that has traits of both obsession and bullying. Furthermore, offenders blame and punish targets in the form of stalking. Just as the main cause of cyberbullying is retaliation and punishment against someone who has done wrong or harmed the perpetrator by spreading false rumors about them ([42]; [69]), stalking is also aimed at punishing and harassing the target, because cyber stalkers think victims deserve punishment and harassment. Offenders justify their actions through these processes, which also serve as a major explanatory factor ([21]; [40]; [58]).

Bullying and retribution in stalking tend to occur more often in women than in men ([54]). Similar to the statement that bullying is more prevalent among women than men, the forms of warning and punishment against those who stand out could be more likely to be suitable in cases of female stalking. Just as women who stalk tend to justify their actions by blaming the person who provides the cause ([36]), it takes the form of bullying against female peers to punish and deny out of hate, which is quite different from male stalking. In other words, the motive for punishment, such as denying and blaming the other, can explain female stalking better in the form of bullying. [8] ([8]) suggested that the justification belief of violence in explaining cyber dating abuse was more significant in women and acted as a form of self-defense.

### 2.2. Gender Differences in Control and Opportunity Factors

#### 2.2.1. Control and Opportunity Factors

While the risk factor has a positive (+) relationship with cyberstalking, the control factor has a negative (−) relationship. Here, rather than the motive and risk of cyberstalking, we focus on the factors that can control and prevent it. The most representative theory on the control factor is social control theory ([29]). This theory presents several social bonding factors as an answer to the question of why people do not commit crimes rather than why they commit crimes. In the context of cybercrime, the social bonds assume the individual moral attitude, and ethical sense as moral belief is also important ([35]). In cyberspace, cyberstalking is more permissible because of the characteristics of anonymity and noncontact, so a lack of control over former traits makes people commit cyberstalking more often. In that sense, possessing a moral attitude toward cyberstalking can be a major factor in controlling and preventing stalking. In a stalking study of college students, [24] ([24]) suggested that the attitude toward stalking was the main explanatory factor for both the perpetration and victimization of stalking. In their cyberstalking analysis, [38] ([38]) also suggested that a moral attitude toward stalking was the main factor of cyberstalking. Therefore, individual morality is a major control factor in cases of stalking.

Self-control is another major control factor found to be significant in the previous studies. The general theory of crime by [26] ([26]) noted that most crimes occur immediately and impulsively, so that one’s criminal behavior is differentiated by internal disposition, implying an individual’s capacity to manage and control immediate gratification and impulsivity. Self-control would account for the outbreak of any kind of crime, and even cybercrimes, including cyberstalking, can be explained by a main factor of self-control. In research on stalking ([17]), self-control is evaluated as one of the most important factors in the application of similar self-regulation theory ([5]). In other words, it is easy to act accidentally in cyberspace, and a person with low self-control can easily commit cyberstalking, but if a person with high self-control can refrain from and control their behavior, it can be seen that the possibility of violation is low.

On the other hand, in other discussions on crime, the opportunity factor has been emphasized. For example, the lifestyle theory ([28]) and the routine activity theory ([13]; [48]) emphasize that there should be an opportunity for crime to occur anyway. Unlike the control factor that has a negative (−) relationship with crime, opportunity is a factor that promotes crime and thus is positively (+) related to it. These discussions can also be applied to cybercrime, suggesting that the more opportunities and conveniences there are for crime to occur, the more cybercrime would occur unaffectedly ([30]; [44]). In cyberspace, it is easy to commit a crime with a single click, and there are many accidental opportunities, such as the absence of surveillance, which can be seen as a major cause of crime.

Opportunity factors, such as non-face-to-face (or “untact”) and anonymous situations, can act as major factors in cybercrime ([3]; [41]; [68]). Here, attention is paid to the anonymity of the many opportunity factors. In the case of cyberstalking, anonymity is a major explanatory factor ([53]). [7] ([7]) also finds that, in addition to disinhibition emerging from anonymity, the ease of opportunity and technical ability are other major factors of cyberstalking. As such, opportunity is another major factor that facilitates cyberstalking.

#### 2.2.2. Gender Differences in Their Moderating Effects

Apart from the risk factors, cyberstalking would be more easily explained when control factors are withheld or opportunity factors are disclosed. As explained above, the control factor acts (works) negatively (−) on cyberbullying, whereas the opportunity factor acts (works) positively (+). Such control and opportunity factors can work as a buffer or catalyst integrally in the action of risk factors.

This argument is based on the existing integrated theory. [1]’s ([1]) general strain theory posits that, while strain is a major risk, crime is more likely to occur when social or internal control is low, covering another factor in building up an integrated theory. To some extent, this type of integrated model has been supported in subsequent studies ([45]; [51]). In addition, according to the coercion and social support theory of [14] ([14]), both coercion and supporting factors need to be considered integrally when we divide explanatory factors into two parts: coercion and support. Referencing a study where morality and self-control were utilized as the main factors of control ([4]), [39] ([39]) explained cyberbullying with the main effects and the interaction effects between risk factors and control factors, where the former utilized cyberbullying victimization and cyberbullying peers, while the latter used morality and self-control.

In previous studies, morality and self-control were found to be significant in both genders, showing no difference between men and women in inhibiting crime ([11]; [31]; [59]; [66]). However, when we considered the role of gender difference in the buffering effect of control factors, they appeared to be more significant among women, even if there were not many supporting materials. [34] ([34]) compared the gender effect of strain on crime and suggested that religiosity had a buffering effect on crime in women; this finding was supported by [33] ([33]). This implies that women are less likely to commit crimes if they have the morality to inhibit criminal activity. In addition, [47] ([47]) examined the gender-differentiated buffering effect of morality in the relationship between delinquent peers and crime, and suggested that morality did not buffer the influence of delinquent peers on crime for men, while it played a buffering role for women. This implies that the probability of a criminal act declines with a high level of morality, even if youth have many delinquent peers. Furthermore, the moderating effect of self-control as a controlling factor in the relationship between delinquent peers and crime was greater for women than for men. [12] ([12]) showed that the interaction effect of self-control on the influence of strain on delinquency was greater for women than for men. In other words, these outcomes imply that men are unable to control their morality or self-control in the action of a certain risk, but women are more likely to be able to control their morality or self-control in the process.

On the other hand, it is also likely that the risk factors of cyberstalking would work together with the opportunity factors. In fact, scholars who emphasize opportunity factors have argued that motivated offenders commit crimes when there are opportunities for crime, such as suitable targets and the absence of guardians working together ([13]; [48]). As the opportunity factor used to be treated as a major factor interacting with various risk factors in previous cybercrime studies ([37]), it is likely that criminal opportunity factors would interact with risk factors in explaining cyberstalking. In other words, in the risk factors mechanism, there should be an opportunity for risk to be activated more, and such opportunity factors would promote the impact of risk factors on deviant behaviors.

Some studies on opportunity factors by gender show that opportunity operates more effectively for men than for women. [2] ([2]) argued that women do not easily engage in deviant behavior, even by chance, due to parental control or internal restraints, whereas men have a high plausibility of involvement in delinquency if opportunities arise, due to the manifestation of masculinity in certain circumstances. In addition, [42] ([42]), studying the impacts of various factors on cyberbullying by gender, also suggested that opportunity factors appear to be more important for men. Some research results have shown that factors such as anonymity—a representative variable known as one of the opportunity factors—have a stronger influence on men than on women ([65]). Although few previous studies have dealt with opportunity factors by gender as a moderating variable, we can propose that the effect of opportunity factors as moderators precipitating the impact of risk factors on criminal behavior, such as anonymity, would be greater for men than for women.

## 3. Current Study and Research Questions

To explain cyberstalking, this study proposes a model that focuses on risk, control, and opportunity factors (separately and simultaneously), and empirically tests the effects of these factors on cyberstalking. First, it investigates the main effects of risk factors for cyberstalking. Second, it investigates the interaction effects of risk factors with both control and opportunity factors, respectively, in explaining cyberstalking. Specifically, it is predicted that two types of interaction effects will be tested, such that there will be an interaction effect between the risk factor and the control factor, as well as an interaction effect between the risk factor and the opportunity factor. The former interaction is called suppressing interaction, whereas the latter is termed precipitating interaction.

We used the lack of attachment and the denial of victim risk factors, as the main types and motives of the risk factors are obsession and retribution. According to the discussion on attachment as a risk factor, it is likely that a person with attachment confusion will commit stalking because of problems in their personal relationships, such as anger or frustration in the relationship. From the standpoint of the previous research on stalking ([23]), the lack of attachment seems to occur more likely in men who try to control their relationship with the opposite sex or partner. The denial of victim factor refers to denying and blaming the victim for providing the reasons for stalking, as well as chastising the victim, which seems to be more evident for female stalking than for male stalking ([8]; [36]).

**Research Question** **1.**
*Does lack of attachment and denial of victim affect cyberstalking? Are there gender differences in the impact of these factors?*


Lack of attachment and denial of victim are the main risk factors for cyberstalking, but we posit that stalking would be more likely to occur when control factors such as morality and self-control are low, and less likely to occur when the control factors are high. In the same manner, we posit that the effects of the risk factor on stalking would contribute more to cyberstalking when the opportunity factors are high. In other words, the lack of attachment and denial of victim as risk factors will interact negatively (−) with morality and self-control as control factors, while interacting positively (+) with anonymity as an opportunity factor. Thus, based on previous studies ([33]; [42]; [47]; [65]), it can be assumed that there will be differences between men and women in the moderating effect of the control and opportunity factors in such a way that the former will be greater for women, while the latter will be greater for men.

**Research Question** **2.**
*Do risk factors (lack of attachment and denial of victim) and control factors (morality and self-control) have negative (−) interaction effects with cyberstalking? Are there gender differences in the impact of these factors?*


**Research Question** **3.**
*Do risk factors (lack of attachment and denial of victim) and opportunity factors (anonymity) have positive (+) interaction effects on cyberstalking? Are there gender differences in the impact of these factors?*


## 4. Data and Methods

### 4.1. Sample and Procedure

In this study, we examined SNS cyberstalking among adult college students. For the survey, we selected 60 students from each of the five universities in Seoul, South Korea. One university was selected from each of five regions in Seoul: east, west, north, south, and central. A quota sampling was applied within each university with three demographic variables: gender, academic major, and academic level. The Survey was conducted over two weeks in July 2018, with approval from the Institutional Review Board of the first author’s institution (IRB#SSU-202503-HR-626-1). Participants were informed that their participation was completely voluntary, and their responses would be treated as confidential and protected by the law. A total of 304 male and female students participated in the survey. Among them, individuals who responded “yes” to the question regarding SNS usage were selected, resulting in a final sample of 270 SNS users for analysis.

We conducted OLS regression analysis using SPSS version 24. The baseline model examined the effect of two risk factors on cyberstalking by gender, controlling for sociodemographic variables. In addition, we incorporated interaction terms with each moderating variable. To mitigate multicollinearity, the interaction terms were mean-centered.

### 4.2. Measurements

We used the actual perpetration of cyberstalking as the dependent variable in the current study. While previous studies have used inconsistent measurements for this variable ([15]; [42], [43]), we specifically defined cyberstalking as more than just accessing or monitoring a partner’s social media account without consent. In our context, it refers to the act of stalking, which includes sending persistent messages to a partner against his/her wishes or causing anxiety and fear. To operationalize this definition, participants were asked to respond to the following questions for the past year, “I have continuously sent messages on SNS even though the victim rejected it”, “I have consistently sent a friend request on SNS even though the victim rejected it”, “I have uploaded a post that could cause fear or anxiety in a victim on SNS”, “I have never sent a message or made a comment that could cause fear or anxiety against the victim on SNS”. The four items have a four-point response scale: “never (score = 0)”, “once or twice (score = 1)”, “three to eight times (score = 2)”, and “nine or more times (score = 3)”. Each score (0–3) of the four items was summed and the natural log-transformation of the summed score was applied to reduce positive skewness in the variable (alpha = 0.960).

Lack of attachment, as a risk factor corresponding to attachment avoidance and attachment anxiety, was constructed using 12 items among the items on the attachment scale developed by [10] ([10]), such as “I feel uncomfortable with being too close to others” and scored on a 5-point Likert scale, ranging from “not at all true (1)” to “very true (5)”. The responses were then summed, resulting in a composite score (alpha = 0.922).

Denial of victim, as a risk factor, utilized items from the discussion of techniques of neutralization ([63]) in stalking situations. Three items were selected from the study by [8] ([8]), such as “Anyone stalked is responsible for the victimization to a degree”, and scored on a 5-point Likert scale (alpha = 0.890).

The morality of cyberstalking, as a control factor to control the impact of the risk factor, consists of one question regarding how bad the cyberstalking behavior is on a 5-point Likert scale from “not bad at all (1)” to “very bad (5)”.

Self-control, based on a previous study ([27]), consisted of two questions for each of the following six traits: impulsivity, risk-seeking, simple task-seeking, activity, selfishness, and temperament. Thus, 12 questions in total, including “I often act impulsively”, were asked to obtain answers on a 5-point scale, ranging from “not at all true (1)” to “very true (5)”. The responses were then summed, resulting in a composite score, and finally were inverse-coded (alpha = 0.823).

Anonymity, as an opportunity factor, was asked to check whether they used SNS mainly in anonymous situations. Four questions were asked: “Actually, I do not use my real name on SNS”, and responses were made on a 5-point scale (alpha = 0.906).

As a sociodemographic control variable, age was recorded based on the year of birth. In addition, the family’s subjective economic status (SES) was assessed using a 5-point Likert scale ranging from “low” to “high”.

## 5. Results

Table 1 presents descriptive statistics of the variables used in this study. The study sample consisted of 145 males (53.7%) and 125 females (46.3%). The average age of the respondents was 21.37, ranging from 18 to 28 years, with an average age of 21.9 years for males and 20.9 for females. The average value of the family’s economic level was 3.18, ranging between 1 and 5, with 3.24 for males and 3.13 for females. The average lack of attachment as an independent variable was 32.452, ranging between 12 and 60, with males scoring 34.064, a little higher than females (31.062). The average denial of victims was 5.024 in the range of 3–15, with males scoring 6.408, which was significantly higher than that of females (4.166). Morality, as a control factor, had an average of 4.390, ranging between 1 and 5, with males scoring 3.944, which was lower than that of females (4.778). The average self-control score was 37.478 in the range of 12–60, and males scored 36.960, which was lower than that of females (37.747). As an opportunity factor, the average of anonymity in the range of 4–20 was 10.048, and males scored 10.912, which was higher than that of females (9.303). Finally, the average logged score of cyberstalking experience, the dependent variable, has an average of 0.512, ranging between 0 and 2.56, with males (0.814) scoring much higher than females (0.252). The gender differences in mean values for all variables were tested and are presented in Table 1.

Prior to the analysis of the major research question, Table 2 present the effects of the risk factors on cyberstalking. In the case of males, as shown in Table 2, lack of attachment was significant with the largest effect size (β = 0.524), followed by denial of victim (β = 0.347), both at the *p* < 0.001 level. In the case of females, in Table 2, denial of victim had the greatest effect (β = 0.315) at the *p* < 0.001 level. However, lack of attachment was not significant for females. When gender was included as a dummy variable (male = 1, female = 0) to examine interaction effects, a significant gender difference was found in the effect of lack of attachment (at the *p* < 0.001 level), which supports the predictions of this study to some extent.

Table 3, Table 4 and Table 5 present the results of the interaction effect between risk, control, and opportunity factors. Table 3 presents the interaction effect between the risk factors and the control factor, morality. The direct and independent influence of morality was significant for men (β = −0.805) at the *p* < 0.001 level, but not for women (β = −0.157). Next, we investigated the interaction effects between risk factors (lack of attachment and denial of victim) and morality. It showed that the interaction effect between lack of attachment and morality was not significant for both men and women, but the interaction effect between denial of victim and morality was significant at the *p* < 0.05 level only for women (β = −0.303) even if it was still not significant for men. In the analysis of gender differences through interaction effects, the results of three interaction effects indicate that the interaction effect between denial of victim and morality is significantly stronger among women (at *p* < 0.05 level). This means that the moderating effect as a control factor for morality works more in the case of women.

Table 4 presents the interaction effect between the risk factors and the control factor, self-control. The direct and independent influence of self-control was not significant for both men and women. Next, we investigated the interaction effects between risk factors (lack of attachment and denial of victim) and self-control. The interaction effects between the risk factors and self-control were not significant for men, but only one interaction effect was significant for women, that is, between denial of victim and self-control (β = −0.211) at the *p* < 0.05 level. In the analysis of gender differences through interaction effects, however, the results of three interaction effects indicate that the interaction effect between denial of victim and self-control shows no significant difference between men and women.

Finally, in Table 5, the interaction effect between the risk factors and the opportunity factor for anonymity is provided. Unlike the previous steps, the direct and independent influence of anonymity on cyberstalking was significant for both men (β = 0.263) and women (β = 0.319) at the *p* < 0.001 level. Next, we investigated the interaction effects between risk factors (lack of attachment and denial of victim) and anonymity (an opportunity factor) were investigated. Results showed that the interaction effect between lack of attachment and anonymity was significant for men (β = 0.251), but it was not significant for women. It showed that the interaction effect between denial of victim and anonymity was not significant for both men and women. In the analysis of gender differences through interaction effects, the results of three interaction effects indicate that the interaction effect between lack of attachment and anonymity is significantly stronger among males (at *p* < 0.01 level). This implies that the moderating effect of anonymity as an opportunity factor works more in the case of men.

## 6. Discussion

To explain cyberstalking in the use of SNS, this study investigated the influence of risk factors for cyberstalking and also tried to test the interaction effects focusing on risk, control, and opportunity factors simultaneously. The proposed hypothesis posits that there will be a negative (−) interaction (risk factor × control factor) and a positive (+) interaction (risk factor × opportunity factor). In this study, we used lack of attachment and denial of victim as risk factors, morality and self-control as control factors, and anonymity as an opportunity factor.

Furthermore, this study predicted that the interaction between risk and moderating factors (control and opportunity) would differ by gender. It posits that the impact of the risk factors—lack of attachment and denial of victim—would be differentiated by gender such that lack of attachment would be important for men and denial of victim for women. In the effect of these risk factors, the interaction effect is also differentiated by gender in such a way that the moderating effect of control factors will be greater for women than for men, and the effect of opportunity factors, such as anonymity, will be greater for men than for women. Thus, the interaction between risk factors (lack of attachment and denial of victim) and control factors (morality and self-control) would be greater in women, while the interaction between risk factors (lack of attachment and denial of victim) and opportunity factors (anonymity) would be greater in men.

The research outcomes support the general hypotheses of this study. In the case of men, stalking originates from the obsession with a former lover of the opposite sex. For men, denial of victim was also significant, but the effect of lack of attachment was stronger. Unlike men, women’s lack of attachment was not significant, and the effect of denial of victim was significant, suggesting that stalking is an extension of cyberbullying, and is committed as a form of punishment to deny or blame the other person. An analysis of gender differences in the effects of the two risk factors revealed a significant gender difference in the effect of lack of attachment.

This study predicted that the moderating role of the control factors would be greater for women than for men. As a result, the interaction effect of denial of victim × morality appeared more significant in women, with a significant gender difference observed. However, lack of attachment × morality was not significant for either men or women. Additionally, the results showed that denial of victim × self-control was significant in women; however, gender difference analysis indicated no significant difference between men and women. This suggests that morality reduced the impact of the denial of victim, a key risk factor for women.

This study predicted that the facilitating role of the opportunity factor would be greater for men than for women. It was shown that the role of opportunity factors in interaction differs between men and women. Specifically, the interaction effect of lack of attachment × anonymity was more significant in men, with a significant difference compared to women, even though denial of victim × anonymity was not significant for either men or women. What it suggests is that, as predicted, the opportunity factors in the interaction effects facilitate and precipitate the action of risk factors on cyberstalking for men.

From these study outcomes, we found that the cause of cyberstalking behavior is somewhat different between men and women, and that the moderating effects of control factors and opportunity factors on cyberstalking through risk factors could also be differentiated by gender. This implies that the action of risk factors in cyberstalking needs to be identified separately for each gender. For men, addressing the problems of fixation and parental nurturing is critical to preventing cyberstalking. This finding is consistent with previous research that emphasized the importance of fixation or attachment style ([23]), For both women and men, retaliation, punishment, and blaming others are important causes of cyberstalking, and the change of the justification belief is necessary. These results imply that gender-specific solutions are required, such as counseling and healing for male stalkers to reduce their obsession with their former partner or lover, and for women to get along and foster relationships with their same-sex friends. Furthermore, a gender-differentiated policy needs to be pursued in search of countermeasures that contribute to controlling the action of the risk. For example, moral and ethics education may be more effective and suitable for women, whereas the removal of opportunities for cyberstalking would be more effective and suitable for men.

Now that the empirical research on cyberstalking is scarce, this study could be considered to be meaningful and contributable for developing the subject in this field in bits and pieces. It is also meaningful to clarify the process by which the moderating action of control and opportunity factors with risk factors on cyberstalking can be differentiated by gender. However, this study assumes that men reported stalking women and women reported stalking their same-sex friends; but, in the future, it will be necessary to study more in-depth through stalking surveys that identify the target-gender of each stalker, including intimate partner stalking ([25]; [55]; [61]). In addition, this study focused on gender differences in stalking motives and risk factors, specifically examining only two factors: lack of attachment and denial of victim. However, future research should consider a broader range of factors to provide a more comprehensive understanding. A limitation of this study is that the sample was drawn exclusively from college students, which may limit the generalizability of the findings. Furthermore, the cross-sectional research design and relatively small sample size pose methodological concerns that could reduce its statistical power. Future studies should adopt longitudinal research designs and include larger, more diverse samples to enhance the robustness of the findings. We also acknowledge that our measure of self-control was not ideal, as we used a shortened version (12 items instead of the original 24) of [27]’s ([27]) low self-control scale. Additionally, while this study addressed the skewness of the dependent variable through log transformation, future research should consider alternative statistical approaches, such as logistic regression or other advanced analytical techniques, to further refine the analysis. While this study offers a preliminary example, future research will need to build a more robust integrative model that includes risk, control, and opportunity factors, and may require the construction of gender-integrated or gender-specific models. Future research should continue to expand the sample by targeting not only college students but teenagers and adults as well.

## 7. Conclusions

This study found that the effects of lack of attachment and denial of victims as risk factors for cyberstalking differed by gender. As a risk factor, the influence of lack of attachment was greater in men, while denial of victim was greater in women. Furthermore, in the context of risk factors, we predicted that the moderating effect of the control factor would be greater for women, and that of the opportunity factor, such as anonymity, would be greater for men. The results indicate that the moderating effects of control factors were stronger for women, as evidenced by the significant interaction between denial of victim and morality. By contrast, the moderating effect of opportunity factors was stronger for men, with the interaction between lack of attachment and anonymity being more significant for men. These findings suggest that the causal processes of cyberstalking and the respective moderating actions can be differentiated across gender, which implies that a gender-differentiated policy needs to be implemented to control the action against cyberstalking.

## Figures and Tables

**Table 1 behavsci-15-00566-t001:** Descriptive Statistics of the Variables (Mean, S.D., and Range).

	Total		Male		Female		Range
	M	SD	M	SD	M	SD	
* Age	21.370	2.047	21.900	2.030	20.900	1.952	18–28
SES	3.180	0.742	3.240	0.807	3.130	0.680	1–5
Lack of Attach	32.452	10.281	34.064	12.058	31.062	8.251	12–60
* Denial of Victim	5.204	2.881	6.408	3.432	4.166	1.740	3–15
* Morality	4.390	1.079	3.944	1.340	4.778	0.548	1–5
Self-control	37.478	7.455	36.960	8.088	37.747	6.858	12–60
* Anonymity	10.048	4.451	10.912	4.750	9.303	4.047	4–20
* Cyberstalking	0.512	0.860	0.814	1.015	0.252	0.591	0–2.56

* Denotes a significant mean difference between males and females (*p* < 0.05 (two-tailed test)).

**Table 2 behavsci-15-00566-t002:** Impact of Risk Factors on Cyberstalking.

			Cyberstalking			
	Male		Female		Total	
	b	β	b	β	b	β
Age	−0.032	−0.064	0.031	0.101	0.031	0.073
SES	−0.040	−0.032	−0.058	−0.067	−0.058	−0.050
Lack of Attach	0.044 ***	0.524	0.009	0.122	0.009	0.105
Denial of Victim	0.102 ***	0.347	0.107 ***	0.315	0.107 **	0.258
Male					0.404	0.135
Age * Male					−0.062	−0.100
SES * Male					0.018	0.036
Lack of Attach * Male					0.035 ***	0.575
Denial of Victim * Male					−0.004	−0.021
Adj R square	0.609		0.118		0.525	
F score	49.381 ***		5.802 ***		34.073 ***	

Notes. * *p* < 0.05, ** *p* < 0.01, *** *p* < 0.001.

**Table 3 behavsci-15-00566-t003:** Multiple Regressions of the interaction effects with morality on Cyberstalking.

			Cyberstalking			
	Male		Female		Total	
	b	β	b	β	b	β
Age	0.012	0.024	0.027	0.052	0.027	0.063
SES	0.029	0.023	−0.055	−0.067	−0.055	0.022
Lack of Attach	0.012 *	0.139	0.007	0.112	0.007	0.082
Denial of Victim	0.012	0.040	0.101 ***	0.330	0.101 ***	0.338
Morality	−0.610 ***	−0.805	−0.123	−0.157	−0.123	−0.154
LoA * Morality	0.002	0.030	−0.000	−0.041	−0.000	−0.008
DoV * Morality	0.004	0.027	−0.056 *	−0.303	−0.056 *	−0.242
Male					−0.276	−0.160
Age * Male					−0.015	−0.190
SES * Male					0.084	0.166
Lack of Attach * Male					0.005	0.105
Denial of Victim * Male					−0.089 *	−0.210
Morality * Male					−0.487 ***	−0.577
LoA * Morality * Male					0.002	0.035
DoV * Morality * Male					0.060 *	0.262
Adj R square	0.698		0.138		0.651	
F score	71.095 ***		4.278 ***		34.394 ***	

Notes. * *p* < 0.05, ** *p* < 0.01, *** *p* < 0.001.

**Table 4 behavsci-15-00566-t004:** Multiple Regressions of the interaction effects with self-control on Cyberstalking.

			Cyberstalking			
	Male		Female		Total	
	b	β	b	β	b	β
Age	−0.032	−0.064	0.034	0.112	0.034	0.081
SES	−0.028	−0.022	−0.059	−0.067	−0.059	−0.051
Lack of Attach	0.044 ***	0.520	0.007	0.093	0.007	0.081
Denial of Victim	0.099 ***	0.335	0.078 **	0.230	0.078 *	0.262
Self−Control	0.000	0.001	−0.016	−0.183	−0.016	−0.137
LoA * Self−Control	0.001	0.052	0.000	0.026	−0.000	−0.022
DoV * Self−Control	−0.002	−0.067	−0.008 *	−0.211	−0.008	−0.208
Male					0.854	0.195
Age * Male					−0.066	−0.144
SES * Male					0.031	0.062
Lack of Attach * Male					0.037 ***	0.511
Denial of Victim * Male					0.021	0.095
Self−Control * Male					−0.016	−0.140
LoA * Self−Control * Male					0.000	−0.026
DoV * Self−Control * Male					−0.006	−0.126
Adj R square	0.602		0.129		0.523	
F score	27.843 ***		3.980 ***		20.413 ***	

Notes. * *p* < 0.05, ** *p* < 0.01, *** *p* < 0.001.

**Table 5 behavsci-15-00566-t005:** Multiple Regressions of the interaction effects with anonymity on Cyberstalking.

			Cyberstalking			
	Male		Female		Total	
	b	β	b	β	b	β
Age	−0.020	−0.040	0.029	0.096	0.29	0.069
SES	−0.084	−0.067	−0.077	−0.089	−0.077	−0.066
Lack of Attach	0.033 ***	0.397	0.003	0.035	0.003	0.030
Denial of Victim	0.045	0.153	0.099 ***	0.291	0.099 ***	0.331
Anonymity	0.056 ***	0.263	0.047 ***	0.319	0.047 **	0.241
LoA * Anonymity	0.005 ***	0.251	0.000	−0.013	0.000	−0.010
DoV * Anonymity	0.001	0.016	0.014	0.204	0.014	0.133
Male					0.331	0.192
Age * Male					−0.049	−0.128
SES * Male					−0.007	−0.013
Lack of Attach * Male					0.031 ***	0.578
Denial of Victim * Male					−0.054	−0.147
Anonymity * Male					0.010	0.071
LoA * Anonymity * Male					0.005 **	0.239
DoV * Anonymity * Male					−0.013	−0.100
Adj R square	0.679		0.155		0.580	
F score	38.532 ***		4.782 ***		25.740 ***	

Notes. * *p* < 0.05, ** *p* < 0.01, *** *p* < 0.001.

## Data Availability

The datasets presented in this article are not readily available due to technical and language limitations. Requests to access the datasets should be directed to the corresponding author.

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
