# Peer review of "Gender Differences in Cyberstalking: The Roles of Risk, Control, and Opportunity Factors in Social Media"

_behavsci, 2025, doi:10.3390/bs15050566_

Round 1

Reviewer 1 Report

Comments and Suggestions for Authors

The authors have studied an important topic (cyberstalking perpetration) and collected data on several interesting factors that could explain cyberstalking perpetrated by college students. I feel they have valuable data but the analyses performed do not match the aims and conclusions (please see my remarks below). Also, I feel the introduction could benefit from restructuring and shortening. Finally, abstract could be improved. Overall, I feel that, in light of these concerns, the manuscript is not eligible for publication in this journal.

Abstract:

  • Abstract: Sentence ‘The effects of lack of attachment and denial of victims as risk factors for cyberstalking differed by gender.’ Seems to refer to results. Please make clear that this is a hypothesis.
  • ‘Moderating actions? => reword
  • Were data cross-sectional? Please mention in abstract
  • Results are not described in a clear manner.

Introduction:

Introduction needs restructuring, for instance:

  • It would be better to integrate the literature review in the part before paragraph 2.1 and shorten the introduction.
  • The end of the paragraph in lines 61-75 (part between lines 69-75) of page 2 is somewhat unclear.
  • In line 111 p. 3 it is not clear which ‘above classification’ is referred to. It is better to repeat it for clarity.
  • Message of paragraph in lines 129-138 p. 3 is not evident.

Method:

  • How many students were approached, the 304 students (or more)?
  • Was the skewness of the dependent variable resolved with the log-transformation (i.e. were assumptions met)? You could also consider performing logistic regression analyses (with a dichotomous dependent variable of 0 = never vs. 1 = once or more times).
  • Good reliabilities of the scales.

Results:

  • Please add test statistics to Table 1 for the gender differences in the independent variables.
  • It is not possible to directly compare the results of the males and females in Table 2 and in Table 3-1 and 3-2, for such a test, the researchers should test for interactions between gender and the other independent variables (at the third step, see below).
  • In Table 3-1, the second B in the first model should be a beta.
  • It is best to put Age and SES at a first step, at the second step the other independent variables, and at the final step the interactions (for each model). Please add the R square change for each step. It is also possible to use a multi-group SEM model (with gender as a grouping variable).

Discussion:

  • Unfortunately, most of the conclusions (about the gender differences) cannot be drawn on the basis of these analyses.
Comments on the Quality of English Language
  • Abstract: Sentence ‘The effects of lack of attachment and denial of victims as risk factors for cyberstalking differed by gender.’ Seems to refer to results. Please make clear that this is a hypothesis.
  • ‘Moderating actions? => reword
  • The end of the paragraph in lines 61-75 (part between lines 69-75) of page 2 is somewhat unclear.
  • Message of paragraph in lines 129-138 p. 3 is not evident.

Author Response

I have revised the paper based on the reviewer's comment. Please take a look at the attached file. 

Reviewer 2 Report

Comments and Suggestions for Authors

page 2 "Is cyberstalking similar to traditional stalking?" this is out of place, probably can just delete it. 

page 3: "It can be said that obsession and affection are not the only motives;
retaliation, punishment, harassment, and collectively in a majority of cases." restate this, it is not very clear what is meant.

page 3: "Thus, we discuss the causal factors of cyberstalking with emphasis on the following two factors." mention what the two factors are before starting the paragraph describing them.

Methods: 

"A total of 304 male and female students were surveyed. Among them, 270 SNS
users were finally selected and analyzed" how did the authors come to the selection of 270 from the 304? was there missing data? did they not fit the selection criteria, what was the selection criteria?

results page 8: "The average value of the family's economic level was 3.18
ranging 1-5 with 3.24 for males and 3.13 for females. " write the actual values of the economic level not just the coded cateogries. what is 3.24 and 3.13? 

For the regression analysis 

four items for the dependent variable, was a factor analysis run? Did these measures actually factor together, what is the results of the reliability analysis?

Instead of running separate models for males and females, run the linear model with gender as a variable and then run gender as an interaction instead of separate models.

avoid using this "On the other hand" it appears several times in the discussion section. 

Limitations section is very limited. There is only a few limitations? Are there any in the way the variables were conceptualized. Was a correlation matrix run to see if any of the variables were highly correlated before putting into the final model? 

Comments on the Quality of English Language

The english language is pretty good. There are just some places where it can be cleaned up grammatically. 

Author Response

(The authors gave the same response as above.)
